# First Report of CTX-M-32 and CTX-M-101 in *Proteus mirabilis* from Zagreb, Croatia

**DOI:** 10.3390/antibiotics14050462

**Published:** 2025-04-30

**Authors:** Branka Bedenić, Josefa Luxner, Gernot Zarfel, Andrea Grisold, Mirela Dobrić, Branka Đuras-Cuculić, Mislav Kasalo, Vesna Bratić, Verena Dobretzberger, Ivan Barišić

**Affiliations:** 1Biomedical Research Center-BIMIS, University of Zagreb, School of Medicine, Department for Clinical Microbiology and Infection Control and Prevention, University Hospital Centre Zagreb, 10000 Zagreb, Croatia; 2Diagnostic and Research Institute of Hygiene, Microbiology and Environmental Medicine, Medical University of Graz, 8010 Graz, Austria; josefa.luxner@medunigraz.at (J.L.); gernot.zarfel@medunigraz.at (G.Z.); andrea.grisold@medunigraz.at (A.G.); 3Department of Anesthesiology, Intensive Medicine and Pain Management, University Hospital Centre Sestre Milosrdnice Croatia, 10000 Zagreb, Croatia; dobric.mirela@gmail.com; 4Department of Microbiology and Hospital Infections, University Hospital Centre Sestre Milosrdnice, 10000 Zagreb, Croatia; branka.gjuras@kbcsm.hr; 5Department of Anesthesiology and Intensive Care, University Hospital Centre Zagreb, 10000 Zagreb, Croatia; kasalomislav@gmail.com (M.K.); vbratic@kbc-zagreb.hr (V.B.); 6Department of Molecular Diagnostics, Austrian Institute for Technology, 1210 Vienna, Austria; verena.dobretzberger@ait.ac.at (V.D.); ivan.barisic@ait.ac.at (I.B.)

**Keywords:** *Proteus mirabilis*, extended-spectrum β-lactamases, plasmid-mediated AmpC β-lactamases, resistance

## Abstract

Background/Objectives: *Proteus mirabilis* is a frequent causative agent of urinary tract and wound infections in community and hospital settings. It develops resistance to expanded-spectrum cephalosporins (ESC) due to the production of extended-spectrum β-lactamases (ESBLs) or plasmid-mediated AmpC β-lactamases (p-AmpC). Here, we report the characteristics of ESBLs and p-AmpC β-lactamases encountered among hospital and community isolates of *P. mirabilis* in two hospitals and the community settings in Zagreb, Croatia. Methods: Antibiotic susceptibility testing was performed using disk-diffusion and broth dilution methods. The double-disk-synergy test (DDST) and inhibitor-based test with clavulanic and cloxacillin were applied to screen for ESBLs and p-AmpC, respectively. PCR investigated the nature of ESBL, carbapenemases, and fluoroquinolone resistance determinants. Selected strains were subjected to molecular analysis of resistance traits by the Inter-Array CarbaResist Kit and whole-genome sequencing (WGS). Results: In total, 39 isolates were analyzed. Twenty-two isolates phenotypically tested positive for p-AmpC and seventeen for ESBLs. AmpC-producing organisms exhibited uniform resistance to amoxicillin-clavulanate, ESC, ciprofloxacin, and sulphamethoxazole-trimethoprim, and uniform susceptibility to carbapenems and piperacillin-tazobactam and all harbored *bla*_CMY-16_ genes. ESBL-positive isolates demonstrated resistance to amoxicillin-clavulanate, cefuroxime, cefotaxime, ceftriaxone, and ciprofloxacin but variable susceptibility to cefepime and aminoglycosides. They possessed *bla*_CTX-M_ genes that belong to cluster 1 (n = 5) or 9 (n = 12), with CTX-M-14 and CTX-M-65 as the dominant allelic variants. Conclusions: The study demonstrated the presence of CTX-M ESBL and CMY-16 p-AmpC among hospital and community-acquired isolates. AmpC-producing isolates showed uniform resistance patterns, whereas ESBL-positive strains had variable degrees of susceptibility/resistance to non-β-lactam antibiotics, resulting in more diverse susceptibility patterns. The study found an accumulation of various resistance determinants among hospital and outpatient isolates, mandating improvement in detecting β-lactamases during routine laboratory work.

## 1. Introduction

*Proteus mirabilis*, a Gram-negative bacterium belonging to the family Morganellaceae, is an important causative agent of urinary tract (UTI) and wound infections in both hospital and community settings. It is mostly isolated from catheter-associated urinary tract infections (CAUTI) associated with the production of biofilms and can further cause bloodstream infections [1]. It develops resistance to expanded-spectrum cephalosporins (ESC) due to the production of extended-spectrum (ESBL) and plasmid-mediated AmpC β-lactamases (p-AmpC) [2].

ESBLs hydrolyze penicillins, first, second, third, and fourth-generation cephalosporins, and monobactams but are in general inhibited by so-called suicide inhibitors such as clavulanic acid, sulbactam, and tazobactam. The first ESBLs were of TEM (Temoniera) and SHV (sulphydryl variable) type. A diversification of ESBLs occurred in the 2000s due to the emergence of the CTX-M (cefotaximase-Munich) family, concomitantly observed in Enterobacterales, most frequently in *Escherichia coli* and *Klebsiella pneumonia* [3]. CTX-M β-lactamases preferentially hydrolyze cefotaxime, have an intrinsic extended-spectrum profile, and are classified into five phylogenetic clusters: CTX-M-1 group, CTX-M-2 group, CTX-M-8 group, CTX-M-9 group, and CTX-M-25 group [4,5]. They are not closely related to TEM or SHV β-lactamases but are typical members of Ambler’s class A and are derived from a gene hosted by *Kluyvera* [4]. In *P. mirabilis*, CTX-M-ESBL variants are also the most common ESBL types today. ESBL-producing *P. mirabilis* has been known since 1999 when TEM and SHV enzymes were still the main responsible enzymes [6]. Plasmids encoding ESBL often carry fluoroquinolone resistance genes *qnr*D and *qnr*S [4,5].

In the late 1980s, cephalosporinase gene (AmpC) of chromosomal origin present in bacteria belonging to the genus Enterobacter, Serratia, Citrobacter, Pseudomonas, and Acinetobacter were identified on plasmids spreading among Enterobacterales without chromosomal ampC gene including *P. mirabilis*. They are Ambler class C enzymes that possess hydrolytic activity against expanded-spectrum cephalosporins (ESC), monobactams, and cephamycins but spare fourth-generation cephalosporins and carbapenems [7]. Unlike ESBLs, they are not susceptible to inhibition with clavulanic acid, sulbactam, or tazobactam but are susceptible to inhibition by cloxacillin and phenylboronic acid [7]. Genes encoding ESBLs and p-AmpC are encoded on plasmids that often contain resistance genes to non-β lactam antibiotics such as aminoglycosides, tetracyclines, sulphonamides, trimethoprim and fluoroquinolones leading to a multidrug-resistant phenotype (MDR). European studies demonstrated the spread of ESBLs and CMY AmpC β-lactamases among *P. mirabilis* isolates in the last decades [8,9].

Previous studies found chromosomal incorporation of the genes encoding AmpC β-lactamases belonging to the CMY (cephamycinase) family, which originates from the chromosomal AmpC gene of *Citrobacter freundii*, and the spread of this resistant clade in Europe without typable plasmids [9]. ESBL-positive isolates were reported as causative agents of an outbreak in a nursing home in Italy [8]. Resistance to aminoglycosides in *P. mirabilis* is due to the acquisition of genes encoding acetylases, adenylases, and phosphorylases (*aacA4*, *aadB*, *aphA6*) [10].

In Croatia, the first official reports date back to 2008 when the first outbreak with ESBL-positive *P. mirabilis* isolates, producing TEM-52, was described in the University Hospital of Split in the southern region of Croatia [11,12]. Later, in 2015, an outbreak of infections associated with a p-AmpC-positive *P. mirabilis*-producing CMY-16 variant from a nursing home in Zagreb was described [13]. P-AmpC-producing isolates demonstrated high-level resistance to ESC and uniform susceptibility to piperacillin-tazobactam, cefepime, and carbapenems. Recently, hospital isolates from the University Hospital of Split were found to possess CMY-16 as well [14]. Haemolysins, urease, motility, and biofilm formation are the virulent factors relevant to UTI [2].

Herein, we analyzed the ESBLs and p-AmpC associated with resistance to ESC in a set of clinical *P. mirabilis* isolates obtained from two tertiary hospitals in Zagreb, which serve both hospital and community settings for the population in Zagreb, in order to investigate dynamic changes in the resistance profiles of this important, but neglected pathogen. Moreover, phenotypic characterization of virulence traits was conducted to measure the potential clinical significance exerted by these isolates. The second goal was to compare different molecular methods for genetic profiling of *P. mirabilis* isolates, such as PCR, whole-genome sequencing (WGS), and the Inter-Array Genotyping CarbaResist Kit method. Genetic profiling using WGS has not been conducted to characterize *P. mirabilis* in Croatia thus far.

## 2. Results

### 2.1. Bacterial Isolates

Thirty-nine isolates were collected in two hospital centers: 38 from University Hospital Centre Zagreb (UHCZ) and one from University Hospital Centre Sestre Milosrdnice (UHCSM). Nine isolates were from hospitalized patients, and the rest were from outpatient settings. In total, 19 isolates originated from urine samples, 11 from wound swabs, two from urinary catheters and sputum, respectively, one from tissue specimens, and the rest from surveillance cultures. Twenty-one females and 18 males were included in the study. The age of the patients ranged from 1 to 91, with a mean value of 69.8 years. There were six patients with UTI, seven mostly males, with CAUTI (prostatic hyperplasia, prostatic adenoma, prostatitis, bladder cancer), two with chronic kidney failure, five with infected decubital ulcers, two with surgical wound infections, and three with malignant disease of haematopoetic system, one with burn infection as shown in Appendix A and Figure 1.

### 2.2. Antimicrobial Susceptibility Testing and Phenotypic Tests for Β-Lactamases

Twenty-two isolates were phenotypically positive for p-AmpC, whereas 17 tested positive for an ESBL, exhibiting augmentation of the inhibition zones around ESC in the presence of clavulanic acid of 10 to 25 mm. AmpC β-lactamases were not inducible.

AmpC-producing organisms were uniformly resistant to amoxicillin alone and combined with clavulanic acid, cefuroxime, ESC (ceftazidime, cefotaxime, ceftriaxone), ciprofloxacin and sulphamethoxazole-trimethoprim and uniformly susceptible to ertapenem, meropenem and piperacillin-tazobactam as shown in Table 1. High resistance rates were observed for gentamicin (95.4%, n = 21) and amikacin (90.9%, n = 20). Cefepime preserved good activity, with six isolates being resistant (27%) and all other being intermediate susceptible or susceptible at increased exposure (73%, n = 16) (Table 1, Appendix A)). All isolates were classified as MDR. Multiple antibiotic resistance indices (MARIs) ranged from 0.46 to 0.6, with a mean value of 0.48. Outpatient isolates prevailed, with only seven isolates (31%) originating from hospitalized patients. There was no difference in the resistance patterns between hospital and community isolates.

Seventeen isolates demonstrated positive double disk-synergy test (DDST) and inhibitor-based tests with clavulanic acid, indicating the production of an ESBL. All isolates exhibited high-level resistance to amoxicillin alone and combined with clavulanic acid, cefotaxime and ceftriaxone with the MIC values of the majority of isolates exceeding 128 mg/L. The best activity was exerted by meropenem with MICs of all isolates in the susceptible range. There were high resistance rates to ceftazidime, cefepime, ciprofloxacin, and gentamicin, with 58%, 94%, 94% and 76% of the isolates showing resistance, respectively. The MARIs ranged from 0.23 to 0.69, with a mean value of 0.61. MICs and the results of phenotypic tests are shown in Appendix A. Similarly, as with AmpC-producing organisms, hospital isolates represented a minority (17%, n = 3) compared to the outpatient setting.

### 2.3. Molecular Detection of Resistance Genes

PCR identified *bla*_CMY_ genes and *bla*_TEM-1_ in all and all but two p-AmpC-positive organisms, respectively. All ESBL-producing organisms were shown to possess *bla*_CTX-M_ and 10 *bla*_TEM_ genes. All except five ESBL-positive isolates were assigned to the phylogenetic group 9, whereas five were allocated to group 1. Eight amplicons were subjected to sequencing (Eurofin, https://eurofinsgenomics.eu/, accessed on 19 December 2024) with forward primer. Blast analysis identified CTX-M-14 encoding genes in four isolates, CTX-M-24 and CTX-M-44, in one isolate—all belonging to the cluster CTX-M-9—and *bla*_CTX-M-15_ genes, belonging to the phylogenetic cluster 1, in two isolates. *bla*_TEM_ genes generated TEM-1. IS*Ecp* element was found upstream of both CTX-M-15 and CMY encoding genes.

### 2.4. Inter-Array Genotyping CarbaResist Method

Two representative isolates were tested, as shown in Table 2. The AmpC-positive isolate contained the *bla*_TEM_ and the *bla*_CMY_ gene associated with the IS*Ecp* insertion element. There were a plethora of aminoglycoside resistance genes encoding acetylases (*aac(*6′)*-Ib* and *aac(*3″)*-Ia*) and adenylases (*aadA1* and *aadA2*), as illustrated in Table 2. Two genes responsible for sulphonamide resistance were identified (*sul*1 and *sul*2) and one for trimethoprim resistance (*dfrA1*) (Table 2).

The ESBL-producing organism was positive for *bla*_CTX-M-9_ as the sole β-lactam resistance determinant. Similarly, as with AmpC-positive organisms, there were a lot of genes associated with aminoglycoside resistance encoding acetylases *(aac(*6′)*-Ib* and *aac(*3″)*-IVa*) and adenylases (*aadA1* and *aadA2*) and *aph* gene for aminoglycoside phosphorylase (Table 2). Identical resistance genes, as in the AmpC-positive organism, were found for sulphonamide resistance (*sul*1 and *sul*2) and trimethoprim resistance, but different allelic variants of the gene for dihydrofolate reductase (*dfr*A5).

### 2.5. Whole-Genome Sequencing (WGS)

Five representative AmpC and nine ESBL-positive isolates were subjected to testing.

All five tested AmpC-positive organisms harbored *bla*_CMY-16_ and all but two *bla*_TEM-1b_. There was a plethora of aminoglycoside resistance genes encoding acetylases (*aac(6′)-Ib3*), adenylases (*aadA1, aadA2*) and phosphorylases (*aph(6)-Id, aph(3″)-Ib, aph(3′)-Ia*) and 16S metylase genes (*arm*A) conferring panaminoglycoside resistance (Table 3). Two allelic variants of sulphonamide resistance genes were found: *sul*1 and *sul*2, as well as *dfrA1* and *dfrA12*, which encode dihydrofolate reductase responsible for trimethoprim resistance. *CatA1* allelic variant of the *cat* gene accountable for the production of chloramphenicol acetyltransferase was found in four out of five sequenced AmpC-producing organisms, as shown in Table 3.

Nine WGS-analyzed ESBL-producing organisms carried various *bla*_CTX-M_ genes—*bla*_CTX-M-14-b_ and *bla*_CTX-M-65_ were found in three isolates, respectively, *bla*_CTX-M-101_ in two, and *bla*_CTX-M-32_ in only one strain. In addition, *bla*_TEM-1d_ was identified in all CTX-M-14 positive isolates, as illustrated in Table 3. Furthermore, *bla*_OXA-1_ was an additional β-lactamase gene to *bla*_CTX-M-65_. There were a lot of different acetyltransferases encoding aminoglycoside resistance genes present, including *aac*(3)-*IIa*, *aac*(3)-*IId*, and *aac*(3)-*IV*, which were found in four, two, and three isolates, respectively. The *aadA1* gene was found in five isolates and the *aadA2* gene in three isolates, generating adenyltransferases, enzymes that modify aminoglycosides. The *aph*(3″)*-Ib* gene encoding aminoglycoside phosphorylases and *aph*(6)*-Id* were present in all isolates, as shown in Table 3. *Cat* and/or *catA1* genes, harbored by all but one strain, which is *catB3* positive, were responsible for chloramphenicol acetyltransferase, while *sul*1 and *sul*2, which mediated resistance to sulphonamides, were found simultaneously in six isolates. Three isolates harbored only *sul*2. Dihydrofolate reductase, rendering trimethoprim inactive, was encoded by three variants of genes: *dfrA1* as the most prevalent and present in six, *dfrA32* in three, and *dfrA17* in two isolates (Table 3). There were three allelic variants of chloramphenicol acetyltransferase genes: *cat*, *catA1*, and *catB3*. *dfrA32* and *catB3* were linked specifically to *bla*_CTX-M-65_ genes. Despite the uniform fluoroquinolone resistance, only one isolate positive for CTX-M-32 carried the plasmid-mediated *qnr*D gene. The *aac*(6′)*-Ib-cr* gene, conferring combined high-level aminoglycoside and fluoroquinolone resistance, was identified in all three of the ESBL isolates harboring *bla*_CTX-M-65._ In contrast to the wide variety of aminoglycoside resistance genes widely distributed among all isolates, there was one tetracycline resistance determinant present in all isolates- *tet(J)*. *Tet(C)* and *tet(H)* allelic variants were present in three and one strains positive for the *bla*_CTX-M-65_ gene, respectively (Table 3).

### 2.6. Conjugation

Conjugation experiments failed to transfer either cefotaxime or cefoxitin resistance to either of the two *E. coli* recipient strains.

### 2.7. Plasmid Analysis by PCR-Based Replicon Typing (PBRT)

No typable plasmids were demonstrated among AmpC- and ESBL-producing *P. mirabilis* isolates by multiplex PCR. However, WGS identified IncCol and IncQ1 plasmids in all tested AmpC-producing organisms and IncQ1 in all tested ESBL-positive isolates.

### 2.8. Detection of Virulence Determinants

All isolates were positive for urease activity, hemolysis, and motility.

### 2.9. Phylogenetic Analysis

A phylogenetic tree was calculated based on WGS data, and it was revealed that all tested AmpC producers belonged to one closely related cluster, with one strain being a singleton (AmpC 4). On the contrary, the ESBL-producing organisms belonged to four different clusters (Figure 1). The ESBL-positive isolates from 2008 (S6_320736 and S2_320735) belonged to a different lineage compared to the ESBL-positive isolates from this study (Figure 2).

## 3. Discussion

The study characterized CTX-M- ESBLs and CMY-16 p-AmpC β-lactamases among hospital and community-acquired isolates in Croatia. The community setting is the major source of MDR *P. mirabilis,* which occurs only sporadically in hospitals. AmpC-producing isolates showed uniform resistance patterns and resistance gene content, whereas ESBL-positive strains had a variable degree of susceptibility/resistance to non-β-lactam antibiotics, resulting in more diverse susceptibility patterns and very diverse resistance genes. The lack of inducibility is consistent with CMY variants, which are deprived of the *amp*R gene upstream of the *bla*_ampC_ gene. CMY-16 detected in p-AmpC isolates is an identical variant already described in the nursing home in Croatia [13]. Later studies demonstrated the diffusion of this important determinant in the hospital setting in the southern region of Croatia [14]. The *ISE*cp insertion element upstream of the *bla*_CMY_ genes increases the expression of the genes and the level of resistance; it is probably responsible for the mobilization event. The fact that cefoxitin resistance was not transferable indicates the possibility that there was chromosomal insertion of *bla*_CMY_ genes, as reported in previous studies [9]. TEM-1 was identified as an additional broad-spectrum β-lactamase among p-AmpC and some ESBL-producing organisms and attributed to increased amoxycillin resistance. This is in concordance with earlier reports from Croatia and Europe, which also found TEM-1 alongside CMY p-AmpC [9,13]. In the previous report from Croatia, AmpC-producing organisms were recovered predominantly from urinary catheters in the nursing home, but in this study, most of the isolates originated from mid-stream urine samples of the outpatient population.

In contrast to ESBL, Amp-C-producing organisms showed almost identical resistance phenotypes with uniform resistance to ESC, aminoglycosides, sulphonamides, and fluoroquinolones, and unform susceptibility to piperacillin-tazobactam and carbapenems, with only slight variations in MIC values and the same resistance gene content. Cefepime is not hydrolyzed by p-AmpC, but some isolates were resistant or intermediate susceptible (increased exposure). This could be attributed to other resistance mechanisms, such as porin loss or hyperexpression of the efflux pump, but the clarification of these resistance mechanisms was beyond this study. In a study conducted in Egypt, the coproduction of p-AmpC and ESBL belonging to the CTX-M family was identified in 60% of the isolates [15]. In their study, VIM-1 (Verona-integron-associated metallo-β-lactamase) and VIM-2 metallo-β-lactamases associated with carbapenem resistance were identified as well [15]. In the present study, the isolates possessed either ESBL or p-AmpC, and none of them exhibited carbapenem resistance. Regarding ESBLs, the shift from TEM-52 reported in the late 2000s [11,12] to the absolute dominance of CTX-M variants was observed. In Italian hospitals, TEM-52 was found to be the most prevalent variant in the past [16].

CTX-M variants reported among *P. mirabilis* isolates belonged predominantly to the CTX-M-9 group, with CTX-M-14 and CTX-M-65 as the dominant variants. This is in contrast with previous studies conducted on *E. coli* and *K. pneumoniae* in which CTX-M-15 was produced by the vast majority of the tested isolates with only sporadic occurrences of CTX-M-3 [17,18,19,20,21]. In the present study, we identified CTX-M-15 allelic variant in two isolates by PCR and sequencing. In this study, a huge number of CTX-M allelic variants was reported, in addition to a plethora of various non-β-lactam resistance genes. CTX-M-101 is the new allelic variant that has not been reported in Croatia thus far but was identified among animal isolates of *E. coli* in South Korea, supporting the One Health approach [22]. Similarly, as the animal isolates, our strains exhibited resistance to all cephalosporins, aminoglycosides, fluoroquinolones, sulphonamides, and trimethoprim. The *ISE*cp insertion element was found upstream of the *bla*_CTX-M-15_ genes. It is responsible for the mobilization of the genes and increases the expression of the gene and the level of resistance [23]. *IS*26 was not found. European reports on ESBLs among *Proteus* spp. are scarce, and the majority of bibliographic references originate from East Asia.

CTX-M-14 was found by PCR and sequencing in three isolates. CTX-M-14b was identified by WGS in three isolates belonging to the phylogenetic cluster 9. These variants were usually reported from animal sources, but there are sporadic reports of human infections, usually from the Far East. The isolate associated with bloodstream infection in China harbored 15 antibiotic resistance genes, including *cat*, *tet(J)*, *bla*_CTX-M-14_ (three copies), *aph*(3′)*-Ia*, *qnr*B4, *bla*_DHA-1_, *qac*E, *sul*1, *arm*A, *msr*(E), *mph*(E), *aadA1*, and *dfrA1*. Aminoglycoside and trimethoprim resistance genes were embedded in class 2 integron [24]. The other report from China described CTX-M-140 as a new variant of CTX-M-14 with decreased hydrolytic activity against cephalosporins [25]. Our isolates exhibited very high MICs of cefotaxime, indicating the enzyme’s high hydrolytic activity. Moreover, CTX-M-14 was frequently isolated from human sources in Taiwan [26]. *bla*_CTX-M-14_ gene was a part of class 2 integron.

CTX-M-65, a derivative of CTX-M-14 and a member of cluster 9, was present in three isolates. The isolates producing CTX-M-65 were previously identified in *P. mirabilis* from Russia with additional *bla*_VEB_ gene encoding Vietnam-extended-spectrum β-lactamase (VEB), *aac*6*-Ib* gene encoding aminoglycoside resistance and *qnrA1* for fluoroquinolone resistance [27]. WGS revealed that the isolates belonged to two different clones. Our CTX-M-65-producing organisms harbored *aac*3*-IIa* and no plasmid-mediated fluoroquinolone resistance determinants. The same allelic variant was identified in animal *P. mirabilis* isolates from Hong Kong. The gene was located in a *Tn*7-like composite transposon and was associated with an extensively drug-resistant phenotype [28]. Unlike previous studies, the ESBL-encoding genes were chromosomally encoded. Genes responsible for sulphonamide resistance *sul*1 and *sul*2, as well as chloramphenicol *catB3*, were located on the chromosome. CTX-M-65 was also described as an allelic variant present in pigs.

CTX-M-32, related to CTX-M-15 and a member of CTX-M-group 1, was detected in only one isolate. This allelic variant was previously identified only in *E. coli*, mostly from chicken and pigs [29,30,31]. To our knowledge, this is the first report of this allelic variant in *P. mirabilis*. CTX-M-24 and CTX-M-44 were registered only sporadically, each in one isolate.

The CTX-M-producing organisms exhibited very variable levels of resistance to ESC, which could be attributed to the differences in plasmid or gene copy numbers or mutations in the promotor region of the gene. The unique characteristic of all CTX-M-producing isolates was the high level of resistance to cefotaxime, but with MICs of ceftazidime, and cefepime being in the susceptible, intermediate, or resistant category. Unlike other Enterobacterales (mainly *K. pneumoniae* and *E. coli*), which were analyzed in our previous studies and were harboring CTX-M-15 associated with high-level resistance to all ESC and cefepime, the *bla*_CTX-M_ genes in the present study belonged predominantly to the phylogenetic cluster 9, and were linked to very variable ESC resistance phenotypes. In contrast to p-AmpC β-lactamases in which CMY-16 remained the only allelic variant for a prolonged time, a significant diversity of the encountered ESBLs was noticed. High-level β-lactam and aminoglycoside resistance was in line with a plethora of various resistance genes.

The fact that no typable plasmids were found by PBRT among AmpC-producing organisms supports the hypothesis of the chromosomal incorporation of *bla*_CMY_ genes. Cefoxitin resistance was not transferable, coupled with negative PBRT, which could indicate the chromosomal location of *bla*_CMY_ genes mediated by IS*Ecp*, as reported by other authors [9]. WGS found IncCol and IncQ1 genes, but since transconjugants were not obtained, we could not prove that they carried either *bla*_CMY_ or *bla*_CTX-M_ genes. This is unusual for *bla*_CTX-M_ genes, which are usually carried by IncFIA, IncW, or IncFIB plasmids. The IncQ1 plasmid found by WGS was not previously identified to spread CTX-M encoding genes; thus, its presence is not necessarily linked to *bla*_ESBL_ genes. The fact that WGS identified plasmids in PBRT-negative isolates points out the advantage of using new molecular techniques in analyzing the molecular epidemiology of resistant isolates.

ESBLs are routinely detected in microbiology laboratories by DDST. However, there is no testing for p-AmpC in the frames of routine microbiology diagnostic, which enables the spread of isolates harboring this important resistance determinant. Failure to detect these enzymes may be responsible for the lack of appropriate infection control measures to prevent the rapid dissemination of resistant pathogens among patients who receive inappropriate therapy. Despite that, we have not observed rapid dissemination of these isolates in the two hospital centers involved in the study. On the contrary, they were more prevalent in the outpatient setting. This is in contrast with the results from Southern Croatia, in which CMY-16 was described among hospital isolates [14].

The phenotypic tests proved very reliable for the detection of both p-AmpC and ESBLs and were in concordance with molecular testing. However, the reporting of AmpC phenotype is underestimated in our laboratory in spite of its diffusion capacity among *P*. *mirabilis.* Inhibitor-based tests with cloxacillin are not routinely conducted, and the results are communicated to clinicians as non-ESBL resistance. Therefore, infection control measures are not recommended in contrast to the ESBL phenotype. MAR indices were lower in AmpC-positive organisms, indicating a smaller number of antibiotics effective against AmpC-positive compared to ESBL-positive strains. Interestingly, cefepime lost activity against some CMY-positive organisms, probably due to hyperproduction of the AmpC enzyme. UTI and wound infections were the predominant sources of MDR *P. mirabilis*. The presence of ESBLs and p-AmpC complicates antibiotic therapy as the therapeutic options are very limited, particularly among oral antibiotics. Sulphamethoxazole-trimethoprim and ciprofloxacin were ineffective in all, and all but one isolate. Co-resistance with non-β-lactam antibiotics, including aminoglycosides and fluoroquinolones, was detected in the majority of isolates. Around half of the isolates with CMY-16 were from urine, mostly from male patients with complicated UTIs, and the other half from wound swabs of the patients with decubital ulcers and other types of wounds. Three patients had malignant haematological diseases, leading to impaired immunity. All CMY-16 positive isolates were susceptible to carbapenems and the majority to cefepime, but both agents are parenteral and not suitable for the outpatient setting. There was no link between the sample type and gender and resistance phenotype. Due to the very small number of isolates from a very large number of different specimens, it was not possible to correlate resistance phenotype and resistance gene content with the sample type. Urinary tract and wounds were the dominant source of MDR *P. mirabilis*. The plasmids encoding ESBLs and p-AmpC often possess resistance genes to non-β-lactam antibiotics, which can also exert selection pressure, enabling the horizontal spread of the plasmids. Only one isolate was shown positive for the plasmid-borne *qnr*D gene, whereas all three CTX-M-65 positive organisms harbored the *aac(6′)-Ib-cr* gene for combined aminoglycoside and fluoroquinolone resistance. In other isolates, fluoroquinolone resistance was probably mediated by mutations of *gyr*A and *par*C genes, as reported by other authors [32]. Still, the investigation of these resistance mechanisms was beyond this study. In Tunis, *qnr*6 was the most frequent fluoroquinolone resistance determinant among ESBL-producing organisms [33].

Urease activity increases the urine pH and enhances the formation of renal stones. Motility and hemolysins are important virulence factors for developing upper urinary tract infections [2].

The limitation of the study is the relatively small number of isolates originating from the same geographic region and the lack of strain typing by established methods such as pulsed-field gel electrophoresis or rep-PCR. However, the strength of the study is the very detailed analysis of the bacterial resistome using different molecular techniques.

## 4. Materials and Methods

### 4.1. Bacterial Isolates

Bacterial isolates were collected in the two largest hospital centers in Zagreb, Croatia, UHCZ and UHSM. The isolates were collected from 2 June 2022 until 19 March 2024 from various clinical specimens and identified at the species level by MALDI-TOF (matrix-assisted laser desorption ionization-time of flight mass spectrometry) Biotyper (Bruker, Daltonik GmbH, Bremen, Germany).

### 4.2. Antimicrobial Susceptibility Testing and Phenotypic Tests for Β-Lactamases

The strains were tested for susceptibility to 13 antimicrobials: amoxicillin alone and combined with clavulanate, piperacillin-tazobactam, cefuroxime, expanded-spectrum cephalosporins (ESCs: ceftazidime, cefotaxime, ceftriaxone), cefepime, imipenem, meropenem, gentamicin, amikacin and ciprofloxacin by broth dilution method according to Clinical Laboratory Standard Institution CLSI [34]. A disk diffusion test, according to the European Committee for Antimicrobial Susceptibility Testing (EUCAST) [35], determined the susceptibility to sulphamethoxazole-trimethoprim, levofloxacin, and chloramphenicol. *E. coli* ATCC 25922 was used as a quality control strain. The intrinsically resistant antibiotics in *Proteus* spp., such as colistin, nitrofurantoin, tigecycline, and tetracycline, were excluded. The isolates were classified as MDR, as described previously by Magiorakos et al. [36]. MARIs were calculated by dividing the sum of antibiotics against which the strain displayed resistance by the total number of antibiotics tested [37]. ESBLs were detected by DDST [38] and confirmed by the combined disk test with ESC and cefepime with and without clavulanic acid, according to CLSI [34]. Augmentation of the inhibition zones around cephalosporin disks by clavulanic acid for at least 5 mm was considered a positive result. Screening for p-AmpC was conducted based on reduced susceptibility to cefoxitin. P-AmpC was confirmed by cefoxitin-cloxacillin DDST (CC-DDST) based on the inhibitory effect of cloxacillin on AmpC enzyme production [39]. A disk of cefoxitin (30 µg) only and another supplemented with cloxacillin solution (20 mg/mL) were placed on Mueller–Hinton agar, inoculated with 0.5 McFarland bacterial suspension, and incubated overnight at 37 °C. An augmentation of the inhibition zone by at least 5 mm by cloxacillin, compared to an unsupplemented disk, was considered a positive result. To study the inducibility of the AmpC enzyme, the cefoxitin-cefotaxime antagonist test was performed as described previously, and the β-lactamase inducibility was confirmed by the presence of a blunted cefotaxime zone adjacent to cefoxitin [40].

### 4.3. Molecular Detection of Resistance Genes

The DNA was extracted by boiling method. Singleplex PCR was applied to detect genes encoding broad and extended-spectrum β-lactamases (*bla*_SHV_, *bla*_TEM_, *bla*_CTX-M,_ *bla*_PER-1_) [41,42,43] and plasmid-mediated fluoroquinolone resistance genes (*qnr*A, *qnr*B, and *qnr*S) [44], was carried out on the isolates phenotypically positive for an ESBL and exhibiting reduced susceptibility to fluoroquinolones, respectively. Multiplex PCR assay was conducted for the detection of p-AmpC on isolates with reduced susceptibility to cefoxitin [45] and to determine five phylogenetic lineages of CTX-M β-lactamases (CTX-M-1, CTX-M-2, CTX-M-8, CTX-M-9, and CTX-M-25) [46], among isolates positive for *bla*_CTX-M_ genes. Isolates positive for *bla*_CTX-M_ and *bla*_CMY_ genes were further tested for the presence of insertion sequence IS*26* and IS*Ecp* by PCR mapping using forward primer for insertion element combined with universal reverse primer for *bla*_CTX-M_ (MA3) and *bla*_CMY_ genes [47]. Eight isolates (P5, P8, P9, P10, P13, P15, P16, and P17) were subjected to sequencing of *bla*_CTX-M_ genes using Eurofin Genomic service Eurofins Genomics (https://eurofinsgenomics.eu/, accessed on 28 April 2025). The positive control strains producing TEM-1, TEM-2, and SHV-1 and SHV-2 were kindly provided by Prof. Adolf Bauernfeind (Max von Pettenkofer Institute, Munich, Germany); CTX-M-15 by Prof. Neil Woodford (Health Protection Agency, London, UK).

### 4.4. Detection of Resistance Genes by Inter-Array Kit CarbaResist

Two *P. mirabilis* isolates (one positive for an ESBL and the other for p-AmpC) were genotyped by an Inter-Array Carba Resist Kit according to the manufacturer’s recommendations (version 1012012100004; INTER-ARRAY, fzmb GmbH, Bad Langensalza, Germany). In brief, genomic DNA was extracted from monoclonal overnight cultures using the Qiagen DNeasy Blood and Tissue Kit (Qiagen, Hilden, Germany) in accordance with the provided manual. The unfragmented DNA was then linearly amplified using one antisense primer per target sequence, with internal labeling via biotin-dUTP incorporation. The resulting single-stranded DNA (ssDNA) products were transferred into ArrayWells for hybridization. Each well contains 230 probes targeting genes encoding key carbapenemases, extended-spectrum β-lactamases (ESBL), AmpC β-lactamases, and other resistance determinants associated with β-lactam, aminoglycoside, fluoroquinolone, sulphonamide, trimethoprim, and colistin resistance. Following hybridization, unbound DNA was removed through washing steps. Horseradish peroxidase (HRP)-conjugated streptavidin then bound to the hybridized biotin-labeled DNA, producing visible dark spots through an enzymatic reaction. These hybridization signals were detected and analyzed automatically using the INTER-VISION Reader system.

### 4.5. Whole-Genome Sequencing (WGS)

Five randomly selected AmpC and nine ESBL-positive isolates were subjected to WGS. For DNA extraction, overnight cultures were grown at 37 °C in Tryptic Soy Broth (Merck Millipore, Boston, MA, USA). Then, the QIAamp DNA Mini Kit (Qiagen, Hilden, Germany) was used according to the manufacturer’s instructions. Subsequently, DNA extracts were brought to the Next Generation Sequencing Facility of the Vienna Biocenter, where MinION (Oxford Nanopore Technology, Boston, MA, USA) was used for sequencing.

The single reads obtained were assembled with Unicycler (https://doi.org/10.1371/journal.pcbi.1005595, accessed on 15 March 2025) and analyzed using the webservers and services of the Center for Genomic Epidemiology (http://www.genomicepidemiology.org, accessed on 15 March 2025) [48]. A phylogenetic tree was generated using the WGS data and the REALPHY online tool (https://doi.org/10.1093/molbev/msu088, accessed on 15 March, 2025). Subsequently, the calculated trees were uploaded to phylo.io (https://doi.org/10.1093/molbev/msw080, accessed on 15 March 2025) for visualization.

### 4.6. Conjugation

The transferability of cefotaxime or cefoxitin resistance was determined by conjugation (broth mating method) employing *E. coli* A15R- strain resistant to rifampicin and *E. coli* J65 resistant to sodium azide [49]. Transconjugants were selected on combined plates containing cefotaxime (2 mg/L) or cefoxitin (8 mg/L) to inhibit the growth of recipient strain and rifampicin (256 mg/L) or sodium azide (100 mg/L) to suppress the donor strains.

### 4.7. Characterization of Plasmids

Plasmids were extracted with a Qiagen Mini kit according to the manufacturer’s instructions. PCR-based replicon typing (PBRT), according to Carattoli et al. [50], and an updated version for the detection of L plasmid [51]. Eighteen pairs of primers were used, including five multiplex and three simplex PCR, to assess the plasmid incompatibility group. Positive control strains were supplied by dr Alessandra Carattoli (Instituto Superiore di Sanita, Rome, Italy).

### 4.8. Detection of Virulence Determinants

Urease activity was determined by inoculating the strains in the urea-containing medium. The change of the color to pink was recorded as a positive result. The hemolytic activity was tested by culturing the strains on a 10% sheep blood plate with the addition of trimethoprim to inhibit swarming. For the motility assay, one colony was stabbed with a one-microliter loop into a semisolid nutrient medium. After overnight incubation at 37 °C, motility was measured as turbidity of the medium [2].

## 5. Conclusions

The study demonstrated the diffusion of β-lactam resistance determinants among hospital and outpatient isolates, mandating improvement in detecting β-lactamases during routine laboratory work. The studies focused on *P. mirabilis* are rare even though it is among the most commonly isolated causative agents of UTI, trailing only behind *E. coli.* P-AmpC still outnumbers ESBL-producing isolates, even though cefoxitin is not licensed in Croatia to exert selection pressure for the diffusion of *bla*_ampC_ genes.

The study did not find any relationship between present isolates and those from 2008, indicating that the present isolates did not evolve from the previous ones but developed de novo.

Our findings underscore significant challenges posed by *P. mirabilis* in terms of antibiotic resistance, with increasing resistance to β-lactam antibiotics due to ESBLs and p-AmpC. These results highlight the severity of *P. mirabilis* as a pathogen and underscore its rapid evolution and adaptability in developing resistance. This study aims to deepen our understanding of the antibiotic resistance mechanisms of *P. mirabilis* to provide important insights for developing future antimicrobial drugs. It promotes effective treatment to bring this pathogen under control and to mitigate its threat to human health.

## Figures and Tables

**Figure 1 antibiotics-14-00462-f001:**
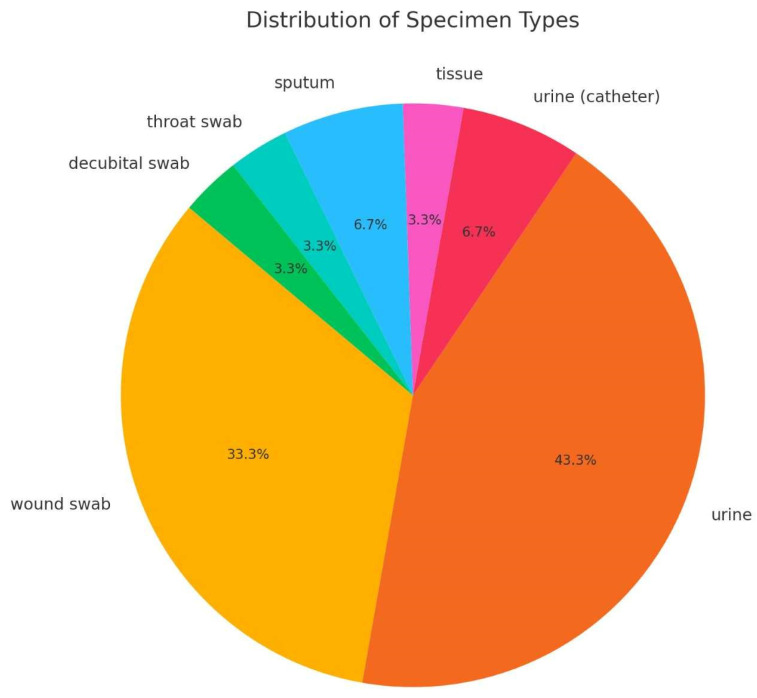
Distribution of isolates according to the specimen type.

**Figure 2 antibiotics-14-00462-f002:**
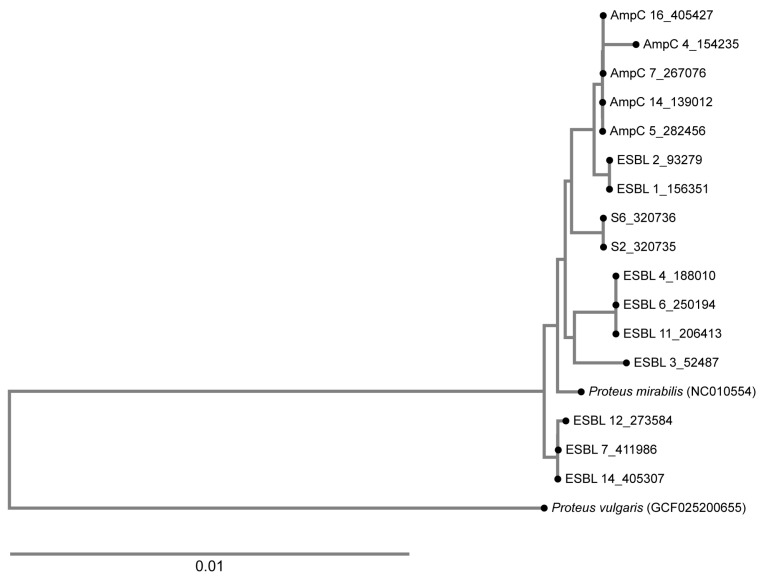
A phylogenetic tree showing the relatedness of the ESBL-producing and the AmpC-producing organisms analyzed in this study. The ESBL isolates belong to four clusters. The isolates from the present study were compared with those from 2008, positive for TEM-52 and designated as S2 and S6, and no relatedness was found. AmpC-producing organisms clustered together, with one strain being a singleton.

**Table 1 antibiotics-14-00462-t001:** Antibiotic susceptibility of ESBL and p-AmpC-producing *P. mirabilis* isolates. MIC values are expressed as mg/L.

	ESBL				AmpC			
	MIC Range	MIC_50_	MIC_90_	Number and % of Resistant Isolates	MIC Range	MIC_50_	MIC_90_	Number and % of Resistant Isolates
amoxycillin-clavulanate	>128–>128	≥128	≥128	17/17 (100%)	>128–>128	≥128	≥128	22/22 (100%)
cefuroxime	>128–>128	≥128	≥128	17/17 (100%)	>128–>128	≥128	≥128	22/22 (100%)
piperacillin-tazobactam	4–32	16	32	0/17 (0%)	2–64	16	64	0/22 (0%)
ceftazidime	2–>128	16	≥128	10/17 (58.8%)	16–>128	>128	>128	22/22 (95)
cefotaxime	32–>128	≥128	≥128	17/17 (100%)	>128–>128	≥128	≥128	22/22 (100%)
ceftriaxone	8–>128	64	≥128	17/17 (100%)	32–>128	≥128	≥128	17/17 (100%)
cefepime	4–64	32	64	16/17 (94%)	4–32	8	32	6/22 (27%)
imipenem	0.5–2	1	2	0/17 (0%)	0.25–1	0.5	1	0/22 (0%)
meropenem	0.06–0.25	0.06	0.12	0/17 (0%)	0.06–0.25	0.12	0.25	0/22 (0%)
gentamicin	0.25–>128	32	>128	13/17 (76.4%)	1–>128	64	>128	21/22 (95.4%)
amikacin	8–>128	32	128	6/17 (35.2%)	32–>128	128	>128	20/22 (91%)
ciprofloxacin	0.25–>128	128	>128	16/17 (94%)	16–>128	128	>128	22/22 (100%)

**Table 2 antibiotics-14-00462-t002:** Inter-Array Chip CarbaResist Kit results of two representative *P. mirabilis* isolates.

Isolate and Protocol Number	β-Lactam	AG	SUL	THR
AmpC 1 (284989)	*bla* _CMY-16_ *bla* _TEM_	*aac(6′)* *aac(3″)-Ia* *aac(6′)-Ib-cr* *aadA1* *aadA2*	*sul*1 *sul*2	*dfrA1*
ESBL 5 (156351)	*bla* _CTX-M-9_	*aac(6′)* *aac(6′)-Ib* *aac(3″)-IVa* *aadA1* *aadA2* *aphA*	*sul*1 *sul*2	*dfrA5*

Abbreviations: AG, aminoglycosides; SUL, sulphonamides; THR, trimethoprim.

**Table 3 antibiotics-14-00462-t003:** Whole-genome sequencing of representative isolates. Accession numbers are provided in a separate section at the end of the manuscript.

Isolate and Protocol Number	β-Lactam	Aminoglycosides	Sulphonamide	Trimethoprim	Chloramphenicol	Tetracycline
AmpC 4 (154235)	*bla* _CMY-16_ *bla* _TEM-1b_	*aac(6′)-Ib3* *aadA1* *aph(6)-Id* *aph(3′)-Ia*	*sul*1 *sul*2	*dfrA1*	*cat* *A1*	*tet(A)* *tet(J)*
AmpC 5 (282456)	*bla* _CMY-16_ *bla* _TEM-1b_	*aac(6′)-Ib3* *aadA1* *aph(6)-Id* *aph(3′)-Ia*	*sul*1 *sul*2	*dfrA1*	*cat* *A1*	*tet(A)* *tet(J)*
AmpC 7 (267076)	*bla* _CMY-16_	*aadA1*		*dfrA1*	*cat*	*tet(J)*
AmpC 14 (139012)	*bla* _CMY-16_	*aadA1*		*dfrA1* *dfrA12*		*tet(A)* *tet(J)*
AmpC 16 (405427)	*bla* _CMY-16_ *bla* _TEM-1b_	*aadA1* *aadA2* *aac(6′)-Ib*3 *aph(6)-Id* *aph(3′)-Ia* *aph(3″)-Ib* *arm*A	*sul*1 *sul*2	*dfrA1* *dfrA12*	*cat* *A1*	*tet(A)* *tet(J)*
ESBL 1 (156351)	*bla* _CTX-M-101_	*aph(6)-Id* *aph(3′)-Ia* *aph*(3″)-*Ib* *aac*(3)-*IId* *aadA1* *aadA5*	*sul*1 *sul*2	*dfrA1* *dfrA17*	*cat* *catA1*	*tet(J)*
ESBL 2 (93279)	*bla* _CTX-M-101_ *bla* _TEM-1d_	*aph(6)-Id* *aph(3′)-Ia* *ap*h(3″)-*Ib* *aac*(3*)-IId* *aadA1* *aadA5*	*sul*1 *sul*2	*dfrA1* *dfrA17*	*cat* *catA1*	*tet(J)*
ESBL 3 (52487)	*bla* _CTX-M-32_	*aph(6)-Id* *aph(3′)-Ia* *aph*(4)*-Ia* *aph*(3″)*-Ib* *aac*(3)*-IV*	*sul*1 *sul*2	*dfrA1*	*cat*	*tet(J)*
ESBL 4 (188010)	*bla* _CTX-M-14b_ *bla* _TEM-1d_	*aph(6)-Id* *aac(*3*)-IIa* *aadA1* *aph(3″)-Ib*	*sul*2	*dfrA1*	*cat*	*tet(J)*
ESBL 6 (250194)	*bla* _CTX-M-14b_ *bla* _TEM-1d_	*aph(6)-Id* *aac(*3*)-IIa* *aadA1* *aph(3″)-Ib*	*sul*2	*dfrA1*	*cat*	*tet(J)*
ESBL 7 (411986)	*bla* _CTX-M-65_ *bla* _OXA-1_	*aph(6)-Id* *aph(3′)-Ia* *aph*(4)*-Ia* *aac(6′)-Ib-cr* *aph(3″)-Ib* *aac*(3*)-IV* *aadA2b*	*sul*1 *sul*2	*dfrA32*	*catB3*	*tet(C)* *tet(J)*
ESBL 11 (206414)	*bla* _CTX-M-14b_ *bla* _TEM-1d_	*aph(6)-Id* *aac(*3*)-IIa* *aadA1* *aph(3″)-Ib*	*sul*2	*dfrA1*	*cat*	*tet(J)*
ESBL 12 (273584)	*bla* _CTX-M-65_ *bla* _OXA-1_ *bla* _TEM-1A_	*aph(6)-Id* *aph(3′)-Ia* *aph*(4)*-Ia* *aac(6′)-Ib-cr* *aph(3″)-Ib* *aac*(3*)-IV* *aadA2b*	*sul*1 *sul*2	*dfrA32*	*catB3*	*tet(C)* *tet*(*H)* *tet(J)*
ESBL 14 (405307)	*bla* _CTX-M-65_ *bla* _OXA-1_ *bla* _TEM-1d_	*aph(6)-Id* *aph(3′)-Ia* *aph*(4)*-Ia* *aac(6′)-Ib-cr* *aph(3″)-Ib* *aac(*3*)-IIa* *aac**(3)-IV* *aadA2b*	*sul*1 *sul*2	*dfrA32*	*cat* *catB3*	*tet(C)* *tet(J)*

## Data Availability

The original contributions presented in the study are included in the article and Appendix A. Further inquiries can be directed to the corresponding author.

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
