# Peer review of "First Report of CTX-M-32 and CTX-M-101 in Proteus mirabilis from Zagreb, Croatia"

_antibiotics, 2025, doi:10.3390/antibiotics14050462_

Round 1
Reviewer 1 Report
Comments and Suggestions for Authors
Dear Author,
This study investigates the presence of extended-spectrum β-lactamases (ESBLs) and plasmid-mediated AmpC β-lactamases (p-AmpC) among both hospital- and community-acquired Proteus mirabilis isolates. The findings highlight the prevalence of CMY-16 and CTX-M variants, addressing an important public health concern. The authors employed robust methodologies and also explored genotype-phenotype correlations. In this regard, I believe the manuscript offers a meaningful contribution to the literature.
However, I would like to suggest a few minor revisions to strengthen the manuscript:
- The study includes only 39 isolates. PBRT results were negative and conjugation experiments were unsuccessful. While this suggests chromosomal integration of blaCMY genes, it would be beneficial to confirm this finding using more definitive techniques such as Southern blotting, if feasible.
- In the discussion section, the authors should elaborate on the unexpected resistance observed in some isolates to antibiotics like cefepime. Possible mechanisms such as efflux pump overexpression or porin loss should be considered and discussed.
- The clinical and epidemiological significance of community-acquired CMY-16–producing isolates should be further emphasized in the discussion.
- Whole-genome sequencing (WGS) data should ideally be visualized using a phylogenetic tree or a gene presence/absence heatmap to enhance clarity and interpretation.
Author Response
Dear Madam/Sir,
Thank you for your valuable comments. We have revised our manuscript according to them.
Dear Author,
This study investigates the presence of extended-spectrum β-lactamases (ESBLs) and plasmid-mediated AmpC β-lactamases (p-AmpC) among both hospital- and community-acquired Proteus mirabilis isolates. The findings highlight the prevalence of CMY-16 and CTX-M variants, addressing an important public health concern. The authors employed robust methodologies and also explored genotype-phenotype correlations. In this regard, I believe the manuscript offers a meaningful contribution to the literature.
However, I would like to suggest a few minor revisions to strengthen the manuscript:
Q. The study includes only 39 isolates. PBRT results were negative and conjugation experiments were unsuccessful. While this suggests chromosomal integration of blaCMY genes, it would be beneficial to confirm this finding using more definitive techniques such as Southern blotting, if feasible.
A. We appreciate your suggestion and we agree, but unfortunately we do not have technical facilities to carry out such experiments.
Q. In the discussion section, the authors should elaborate on the unexpected resistance observed in some isolates to antibiotics like cefepime. Possible mechanisms such as efflux pump overexpression or porin loss should be considered and discussed.
A. We have explained that additional resistance mechanisms such as porin loss or hyperexpression of efflux pumps may be present among cefepime-resistant AmpC producing organisms, but the clarification of this resistance was beyond this study.
Q. The clinical and epidemiological significance of community-acquired CMY-16–producing isolates should be further emphasized in the discussion.
A. We have emphasized the clinical importance of community-acquired CMY-16–producing isolates in the light of very limited therapeutic options. Around half of the isolates were from urine, mostly from male patients with complicated UTI and the other half from wound swabs of the patients with decubital ulcers. Two patients had heamatological malignant disease. All isolates were susceptible to carbapenems and the majority to cefepime, but the both agents are parentheral and not suitable for outpatients setting.
Q. Whole-genome sequencing (WGS) data should ideally be visualized using a phylogenetic tree or a gene presence/absence heatmap to enhance clarity and interpretation.
A. We applied WGS to create a phylogenetic tree already in the first version of the manuscript. Figure 2 is phylogenetic tree. We found that all AmpC producing organisms belonged to one cluster while ESBL positive strains were allocated into four clusters.
Reviewer 2 Report
Comments and Suggestions for Authors
Antimicrobial resistance is one of the emerging global public health threat expected to cause human mortalities exceeding 10 million in the year 2050. Extensive and unregulated use of antibiotics in human and veterinary clinical settings is the major risk factor for AMR . Hence monitoring and characterization of AMR pathogens harbouring AMR genes is paramount important to understand the emerging threat of AMR pathogens and make suitable strategies to mitigate the AMR. In this context the manuscript entitled Diffusion of extended-spectrum and plasmid-mediated Amp-C 2 β-lactamase producing Proteus mirabilis in hospitals and community setting in Zagreb, Croatia, First report of CTX-M-32 and 4 CTX-M-101 in Proteus mirabilis is well focused and the findings “AmpC-producing isolates showed uniform resistance patterns, whereas ESBL-positive strains had variable degrees of susceptibility/resistance to non-β-lactam antibiotics, resulting in more diverse susceptibility patterns is very interesting. Apparently the manuscript does not have any major issues. However the following minor modifications may be carried out before final acceptance
Line no-63- TEM and SHV type- expand the abbreviation
Line no-89- CMY family- expand the abbreviation
Line no-111-118- The major question is does the authors have any data regarding origin of the samples, sex that could link with resistant pattern
Does the authors have any other epidemiological data regarding the resistance pattern?
Line no-130- 0,46 to 0,6 with mean value of 0,48 full stop instead of comma
Line no-138-The MARI ranged from 0,23 to 0,69 with mean value of 0,61. MICs full stop instead of comma
Line no 261- VIM-1 and VIM-2 expand the abbreviation
Author Response
Dear Madam/Sir,
Thank you for your valuable comments. We have revised our manuscript according to them.
Antimicrobial resistance is one of the emerging global public health threat expected to cause human mortalities exceeding 10 million in the year 2050. Extensive and unregulated use of antibiotics in human and veterinary clinical settings is the major risk factor for AMR . Hence monitoring and characterization of AMR pathogens harbouring AMR genes is paramount important to understand the emerging threat of AMR pathogens and make suitable strategies to mitigate the AMR. In this context the manuscript entitled Diffusion of extended-spectrum and plasmid-mediated Amp-C 2 β-lactamase producing Proteus mirabilis in hospitals and community setting in Zagreb, Croatia, First report of CTX-M-32 and 4 CTX-M-101 in Proteus mirabilis is well focused and the findings “AmpC-producing isolates showed uniform resistance patterns, whereas ESBL-positive strains had variable degrees of susceptibility/resistance to non-β-lactam antibiotics, resulting in more diverse susceptibility patterns is very interesting. Apparently the manuscript does not have any major issues. However the following minor modifications may be carried out before final acceptance
Q.Line no-63- TEM and SHV type- expand the abbreviation
A. TEM means Temoniera, the β-lactamase was named after the patients from whom the isolate originated, SHV-sulphydryle variable
Q. Line no-89- CMY family- expand the abbreviation
A.CMY-cephamycinase, AmpC family which originates from the chromosomal AmpC- gene of Citrobacter freundii
Q.Line no-111-118- The major question is does the authors have any data regarding origin of the samples, sex that could link with resistant pattern
A. The data on patients and specimens are provided in the Supplementary Table S1. There was no link between the origin of the sample and gender and resistance phenotype. We have also summarized the epidemiological data in the results section, lines 112-118.
Q.Does the authors have any other epidemiological data regarding the resistance pattern?
A. In the previous studies AmpC-positivity and AmpC resistance pattern was linked to the nursing home residency, but now we do not any have data in our hospital internet system about the stay in the long-term care facility. It was observed that the majority of the urine samples were obtained from the midstream urine samples of the male patients with complicated urinary tract infection (prostatic hyperplasia or adenoma). Infected decubital ulcers and surgical wound infections were also important source of MDR P. mirabilis. There was a small number of strains analyzed from different types of specimens and thus it was not possible to correlate sample type and resistance phenotype. We do not have data if the outpatients were previously treated with broad-spectrum β-lactam antibiotics.
Q. Line no-130- 0,46 to 0,6 with mean value of 0,48 full stop instead of comma
A. corrected
Q.Line no-138-The MARI ranged from 0,23 to 0,69 with mean value of 0,61. MICs full stop instead of comma
A.corrected
Q.Line no 261- VIM-1 and VIM-2 expand the abbreviation
A. VIM-Verona-integron associated metallo-β-lactamase
Reviewer 3 Report
Comments and Suggestions for Authors
This study examines the antibiotic resistance of Proteus mirabilis using disk diffusion and molecular methods. The concept is interesting. However, I have major concerns regarding the study design and the significance of the study. The main finding of the paper is described as: “The study's main finding is the diffusion of CTX-M ESBL and CMY-16 p-AmpC among hospital and community-acquired isolates”. There is no justification for this, and the results do not explain the finding. I, in fact, did not understand what the authors meant by this statement. There is not a single mention of diffusion in the discussion, which is supposed to be an explanation of results. Also, the paper has too few figures, which makes the results a long stretch. There are also too many abbreviations, most of which are not explained. The discussion mentions terms such as “excellent correlation” and “good correlation” but there is not a single p-value in the result to justify this correlation!
The introduction is also difficult to follow. Having connections between sentences would help with this. For instance, line 88-91 need to be explained with respect to relevance to current study. Also, studies in Croatia are mentioned but the significance of current study is not clearly explained, and there is no mention of community acquired infections in the introduction.
Author Response
Thank you for your valuable comments. We have revised our manuscript according to them.
Q.This study examines the antibiotic resistance of Proteus mirabilis using disk diffusion and molecular methods. The concept is interesting. However, I have major concerns regarding the study design and the significance of the study. The main finding of the paper is described as: “The study's main finding is the diffusion of CTX-M ESBL and CMY-16 p-AmpC among hospital and community-acquired isolates”. There is no justification for this, and the results do not explain the finding. I, in fact, did not understand what the authors meant by this statement. There is not a single mention of diffusion in the discussion, which is supposed to be an explanation of results.
A. We have rephrased the sentence: The study characterized CTX-M- ESBLs and CMY-16 p-AmpC β-lactamases among hospital and community-acquired isolates in Croatia. We meant clonal spread by diffusion, as proved by WGS and phylogenetic trees. We found that all AmpC producing organisms clustered togeter except one isolate, whereas there were four clones identified among ESBL producing strains showing high genetic diversity.
Q. Also, the paper has too few figures, which makes the results a long stretch. There are also too many abbreviations, most of which are not explained. The discussion mentions terms such as “excellent correlation” and “good correlation” but there is not a single p-value in the result to justify this correlation!
A. There are two figures provided showing, one with specimen type and the other with phylogenetic tree. By correlation we meant that there was complete concordance between phenotypic and molecular tests, meaning that all isolates phenotypically positive for AmpC yieled CMY- amplicon in PCR and all phenotypically positive for ESBL tested also positive for CTX-M gene in PCR. We understand that the term correlation implies statistical analysis and thus it has been replaced by concordance. The abbreviations are explained: TEM-Temoniera; SHV-sulphydryl variable, MIC-minimum inhibitory concentration; ESC-expanded-spectrum cephalosporins, ESBL-extended-spectrum β-lactamases, p-AmpC-plasmid-mediated AmpC β-lactamases; DDST-double- disk- synergy test, WGS-whole- genome- sequencing; PBRT-PCR- based- replicon typing, UTI-urinary tract infection; CAUTI-catether- associated urinary tract infection
Q. The introduction is also difficult to follow. Having connections between sentences would help with this. For instance, line 88-91 need to be explained with respect to relevance to current study. Also, studies in Croatia are mentioned but the significance of current study is not clearly explained, and there is no mention of community acquired infections in the introduction.
A. The data in lines 88-91 are relevant for the current study because we explained that in the previous studies blaampC genes were incorporated in the chromosome which could explain why in our study conjugation experiments failed and PBRT yielded no typable plasmids. Regarding the significance of the study, we have explained that the study was conducted in the two largest hospitals in Zagreb in order to analyze and compare isolates resistant to expanded-spectrum cephalosporins using new molecular techniques such as WGS. Such studies were not done in Croatia before. It is mentioned that P. mirabilis causes both hospital and community-acquired infections. Genomic characterization of P. mirabilis resistome was not done in our country so far.
Reviewer 4 Report
Comments and Suggestions for Authors
The study presents important findings on antimicrobial resistance in Proteus mirabilis and is valuable for understanding resistance patterns in both hospital and community settings. Addressing the limitations of the study, particularly the small sample size, would enhance the manuscript’s scientific rigor and impact.
The title is informative but could be slightly refined for conciseness, e.g., "First Report of CTX-M-32 and CTX-M-101 in Proteus mirabilis in Zagreb, Croatia."
Line 23 – Remove: “Background/Objectives:”
Line 232 – “The study's main finding is the diffusion of CTX-M ESBL and CMY-16 p-AmpC 232 among hospital and community-acquired isolates in Croatia.” It is better to be cautious with this statement. The sample size is small.
Line 264-265 – Merge this paragraph with the paragraph before.
Conclusion: The conclusion is too long. Focus on providing answers to the study's objectives. Lines 479-487 seem more like a discussion.
The tables are informative, but their readability could be enhanced by reducing redundancy and emphasizing significant results. Especially table 3 title needs to be changed to provide what’s presented in the table.
Figures should be clearly labeled with legends providing sufficient context.
It is necessary to review the standardization of references. Sometimes "doi:XXXXXXXXX" is used, while other times "https://doi.org/10.1111/j.1469-6270691.2011.03570.x" is used.
Author Response
Thank you for your valuable comments. We have revised our manuscript according to them.
Q.The title is informative but could be slightly refined for conciseness, e.g., "First Report of CTX-M-32 and CTX-M-101 in Proteus mirabilis in Zagreb, Croatia."
A.Thank you for your suggestions. We have modified the title as you suggested.
Q.Line 23 – Remove: “Background/Objectives:”
A. We have removed “Background/Objectives“ according to the suggestion, but according to the journal style the abstract has to be structured starting with background objectives.
Q. Line 232 – “The study's main finding is the diffusion of CTX-M ESBL and CMY-16 p-AmpC 232 among hospital and community-acquired isolates in Croatia.” It is better to be cautious with this statement. The sample size is small.
We have removed that sentence and started the discussion section with: The study characterized CTX-M- ESBLs and CMY-16 p-AmpC β-lactamases among hospital and community-acquired isolates in Croatia.
Q. Line 264-265 – Merge this paragraph with the paragraph before.
The two paragraphs have been merged.
Conclusion: The conclusion is too long. Focus on providing answers to the study's objectives. Lines 479-487 seem more like a discussion.
A. The conclusion has been shortened and the paragraph in lines 479-487 has been removed.
The tables are informative, but their readability could be enhanced by reducing redundancy and emphasizing significant results. Especially table 3 title needs to be changed to provide what’s presented in the table.
A. The table has been summarized and the last two columns have been removed to enhance readibility. We consider all the data in the table to be important in order to get a full information on the resistome of the isolates. The NCBI gene accession number have been transferred to a separate section in order to make the table more
Q. Figures should be clearly labeled with legends providing sufficient context.
A. The legend with sufficient context has been added.
Figure 1. Phylogenetic tree showing relatedness of the ESBL and the AmpC- producing organisms analysed in this study. The ESBL isolates belong into four clusters. The isolates from the present study were compared with those from 2008 positive for TEM- 52 and designated as S2 and S6 and no relatedness was found. AmpC producing organisms clustered together with one strain being singleton.
Q. It is necessary to review the standardization of references. Sometimes "doi:XXXXXXXXX" is used, while other times "https://doi.org/10.1111/j.1469-6270691.2011.03570.x" is used.
A. The references have been corrected to meet the journal style.
Round 2
Reviewer 1 Report
Comments and Suggestions for Authors
Dear Author,
I appreciate the thoughtful responses and the revisions made in line with the suggestions. While some limitations remain due to technical constraints, the authors have provided reasonable justifications and have strengthened the discussion accordingly. I find the manuscript suitable for publication in its current form.
Best regards